# UniFlow: Zero-Shot LiDAR Scene Flow via Cross-Domain Generalization

## Abstract

Scene flow estimation is an important primitive for 3D motion understanding and dynamic scene reconstruction. Recent LiDAR-based methods have made significant progress in achieving centimeter-level accuracy on popular autonomous vehicle (AV) datasets. Notably, such methods typically only train and evaluate on the same dataset because each dataset has its own unique sensor setup. Motivated by recent work in zero-shot *image-based* scene flow, we argue that multi-dataset training is essential for scaling up LiDAR-based methods. However, prior work in LiDAR-based semantic segmentation and 3D object detection demonstrate that naively training on multiple datasets yields worse performance than single-dataset models. We re-examine this conventional wisdom in the context of LiDAR-based scene flow. Contrary to popular belief, we find that state-of-the-art scene flow methods greatly benefit from cross-dataset training. We posit that low-level tasks such as motion estimation may be less sensitive to sensor configuration than high-level tasks such as detection. Informed by our analysis, we propose UniFlow, a feedforward model that unifies and trains on multiple large-scale LiDAR scene flow datasets with diverse point density and velocity distributions. Our frustratingly simple solution establishes a new state-of-the-art on Waymo and nuScenes, improving over prior work by 16.4% and 34.5% respectively. Moreover, UniFlow achieves state-of-the-art zero-shot accuracy on TruckScenes, outperforming prior dataset-specific models by 38.4%!

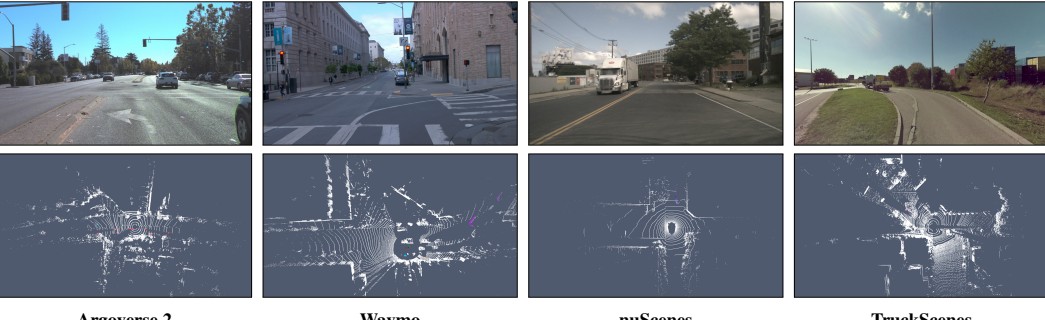

| Argoverse 2 | Waymo | nuScenes | TruckScenes |

Figure 1: **Dataset Diversity.** We visualize the front-center RGB and BEV LiDAR point cloud for Argoverse 2, Waymo, nuScenes and TruckScenes. Notable, all four datasets use different LiDARs, have different sensor configurations and collect data in different environments. Specifically, Argoverse 2, Waymo, and nuScenes collect data in urban city centers with sedans, while TruckScenes primarily collects data on highways with a truck. Due to the diversity of environments and sensor configurations, contemporary LiDAR scene flow methods typically only train and evaluate on the same dataset. However, we find that multi-dataset training significantly improves both in-domain and out-of-domain generalization.

# 1 INTRODUCTION

Contemporary LiDAR-based scene flow methods achieve remarkable performance on popular autonomous vehicle (AV) datasets. However, such methods are often trained and evaluated on the same dataset because each dataset has a different sensor configuration. For example, Argoverse 2 (AV2) (Wilson et al., 2021) uses two out-of-phase 32 beam LiDARs, nuScenes (Caesar et al., 2020) uses one 32 beam sensor, and Waymo Sun et al. (2020) uses a custom sensor. Prior work in LiDAR-based 3D object detection (Wang et al., 2020; Soum-Fontez et al., 2023) and semantic segmentation (Kim et al., 2024; Saltori et al., 2023) shows that naively training models on multiple datasets yields poor performance. Our analysis challenges this conventional wisdom in the context of LiDAR-based scene flow estimation.

**Cross-Domain Generalization for Low-Level Vision Tasks.** FlowNet (Dosovitskiy et al., 2015; Ilg et al., 2017) and RAFT (Teed & Deng, 2020) demonstrate that optical flow models trained on synthetic datasets (e.g. FlyingChairs, FlyingThings3D, Sintel) generalize surprisingly well to casually captured videos. We posit that low-level 3D vision tasks like LiDAR-based scene flow may similarly generalize across different sensors. To test our hypothesis, we train dataset-specific models for AV2 (Wilson et al., 2021), Waymo (Sun et al., 2020) and nuScenes (Caesar et al., 2020) and evaluate zero-shot cross-domain performance (cf. Table 1). Surprisingly, Flow4D (Kim et al., 2025) trained on AV2 and Waymo achieve similar Dynamic Mean EPE across datasets. Notably, Flow4D (Waymo) achieves lower EPE (i.e. better performance) than Flow4D (AV2) on fast moving objects in AV2 since there are more fast movers in Waymo than in AV2. This suggests that cross-domain generalization is highly correlated with training velocity distribution.

**Towards Zero-Shot LiDAR Scene Flow.** Motivated by recent image-based scene flow methods (Liang et al., 2025), we argue that scaling up LiDAR-based methods will be a key enabler for 3D motion understanding and dynamic reconstruction in diverse environments. To this end, we retrain state-of-the-art methods on supervised data from AV2, Waymo, and nuScenes and demonstrate significant improvements for both in-distribution and out-of-distribution generalization. We denote models trained with multiple datasets as UniFlow. To the best of our knowledge, UniFlow is the first to achieve state-of-the-art performance on nuScenes, AV2 and Waymo using a single model. Further, UniFlow demonstrates remarkable zero-shot accuracy on TruckScenes (Fent et al., 2024), outperforming prior dataset-specific models by nearly 40%. Lastly, we propose a test-time refinement step to improve out-of-distribution performance, bridging the gap between supervised and unsupervised scene flow methods. Specifically, we use NSFP (Li et al., 2021) to predict residual flow estimates for UniFlow's predictions, allowing us to trade off test-time inference time for accuracy.

**Contributions.** We present three major contributions. First, we highlight that dataset-specific scene flow models *already* achieve strong performance across datasets, challenging conventional wisdom about LiDAR-based cross-domain generalization. Next, we demonstrate that multi-dataset training yields state-of-the-art performance on AV2, nuScenes, and Waymo, improving over our baselines by an average of 17% across datasets. Lastly, we show that UniFlow achieves state-of-the-art zero-shot performance on TruckScenes, outperforming dataset-specific models by 38.3%.

# 2 RELATED WORK

**Scene Flow Estimation** is the task of describing the 3D motion field between temporally successive point clouds (Vedula et al., 2005; Khatri et al., 2024; Zhang et al., 2025a). Early approaches (Wei et al., 2021; Lang et al., 2023; Wang et al., 2023; Zhang et al., 2024b) learned point-wise features to estimate per-point flow but struggled to scale to large outdoor environments (Zhang et al., 2024a; Vedder et al., 2024). More recent work (Khoche et al., 2025; Kim et al., 2025; Luo et al., 2025; Hoffmann et al., 2025; Vedder et al., 2024) jointly estimates flow for all points in a scene. Contemporary methods can be broadly classified into feedforward models and optimization-based methods. Feedforward models directly learn a mapping between point cloud pairs and flow fields, but require large-scale human annotations (Jund et al., 2021; Kim et al., 2025). For real world datasets (typically from the autonomous vehicle domain), these human annotations are provided in the form of 3D bounding boxes and tracks for every object in the scene. In contrast, optimization-based methods do not require labeled data, and instead optimize a learned representation per-scene (Li et al., 2021; 2023). This per-scene optimization is prohibitively expensive, making it difficult to scale to

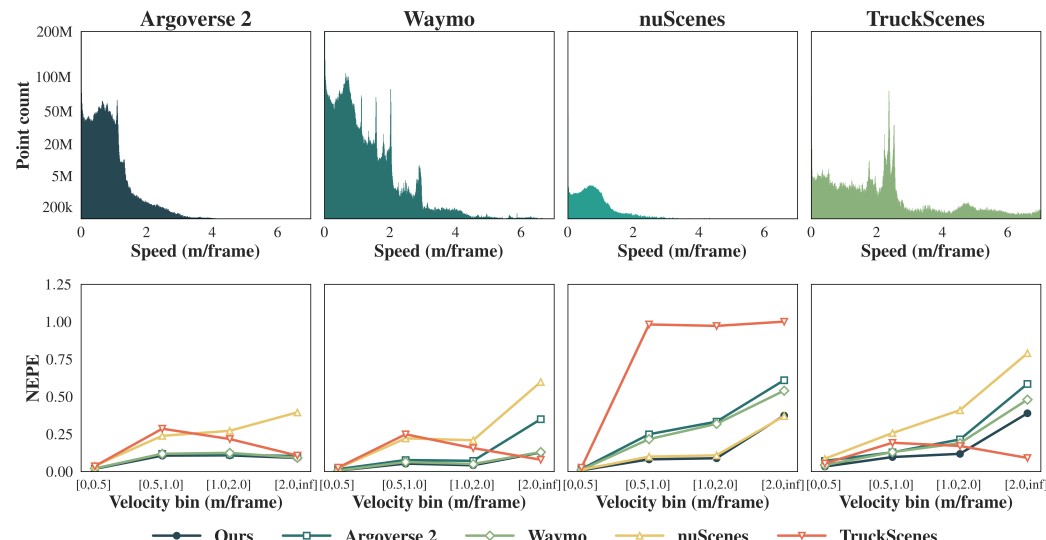

Figure 2: **Cross-Dataset Generalization Correlates with Velocity Distribution.** We plot the velocity distributions for the AV2, Waymo, nuScenes, and TruckScenes train sets (top) and the Dynamic Mean EPE per velocity bin of Flow4D trained on AV2, Waymo, nuScenes, TruckScenes, and UniFlow (bottom). Notably, Flow4D trained on TruckScenes outpeforms Flow4D trained on any other dataset for fast moving objects $(2.0, \infty)$ across all datasets because it has the largest number of fast moving objects.

large datasets. For example, EulerFlow (Vedder et al., 2025) achieves state-of-the-art unsupervised accuracy on many popular AV datasets, but takes over 24 hours to optimize each scene. In summary, feedforward models achieve efficient inference but demonstrate limited generalization beyond their training data, while optimization-based methods produce high quality flow for diverse scenes, but are too slow for real-time applications. Our work aims to reconcile this trade-off by training a single feedforward model that achieves robust generalization across diverse datasets and sensors.

**Cross-Domain LiDAR Generalization** is a long-standing challenge in LiDAR perception. Prior work in 3D object detection Malić et al. (2025); Hegde et al. (2025) and semantic segmentation Kim et al. (2024); Caunes et al. (2025); Xu et al. (2025) shows that models trained on multiple datasets often perform worse than dataset-specific models. This performance drop can be largely attributed to diverse sensor hardware (e.g., number of beams, scan pattern, point density), environmental conditions (e.g., weather, geography), and sensor placement across datasets. To address these challenges, several methods have introduced data augmentation strategies (Sun et al., 2024), such as random point dropping (Wang et al., 2021) and object scaling (Cen et al., 2022), while others (Liu et al., 2024; Michele et al., 2024) have proposed unsupervised domain adaptation techniques that align feature distributions across datasets. However, such data augmentation strategies primarily address the *geometric domain gap* between datasets. For motion-centric tasks such as scene flow, we argue that understanding the distribution of object velocities between datasets represents a critical yet overlooked axis for addressing cross-domain generalization. To this end, our work presents the first systematic study of this *velocity domain gap*.

**Zero-Shot LiDAR Perception** aims to generalize across unseen semantic categories and new LiDAR sensors without retraining. Recent work distills vision-language models from paired RGB-LIDAR data into LiDAR-only models (Ošep et al., 2024; Takmaz et al., 2025; Zhang et al., 2025b; Liang et al., 2025; Khurana et al., 2024; Davidson et al., 2025), facilitating open-vocabulary prompting. However, such methods fail to generalize across different LiDAR sensors. In this work, we introduce UniFlow, a first step toward feedforward LiDAR foundation models for scene flow. We posit that low-level 3D vision tasks like LiDAR-based scene flow generalize more easily across sensors.

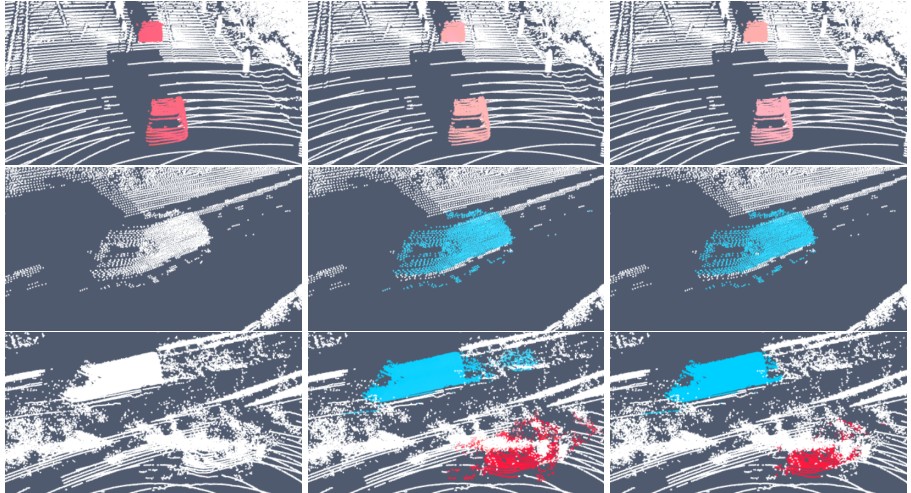

Figure 3: **Zero-Shot Generalization on TruckScenes.** Each row shows one example scene: (a) TruckScenes-Only model, (b) UniFlow model, and (c) ground truth. UniFlow can provide more accurate flows and predict motion where the TruckScenes-only model fails to detect it,

## 3 UNIFLOW: TOWARDS ZERO-SHOT LiDAR SCENE FLOW

In this section, we describe the process of unifying four popular AV datasets for training. Further, we present our test-time refinement network to improve UniFlow's out-of-domain generalization.

**Unifying Datasets.** We standardize both sensor frame rates and annotations across four widely used AV datasets. For example, AV2 and Waymo annotate 3D tracks at 10 Hz, while nuScenes and TruckScenes annotate at 2 Hz. To ensure consistency, we linearly interpolate the nuScenes and TruckScenes tracks to 10 Hz. This step is crucial because state-of-the-art scene flow methods parameterize motion as the displacement between two consecutive LiDAR frames; lower frame rates increase displacement, making scene flow estimation more challenging.

Different from prior work on cross-domain generalization, scene flow is inherently class-agnostic. This allows us to avoid complications arising from dataset-specific label definitions. For example, nuScenes defines `bicycle` as excluding the rider, while Waymo includes the rider (Madan et al., 2023; Robicheaux et al., 2025). Such label ambiguity poses a significant challenge for semantic tasks, potentially explaining why prior LiDAR-based semantic tasks like 3D object detection and semantic segmentation did not benefit from training on multiple datasets.

**Training UniFlow.** We retrain SOTA scene flow methods such as Flow4D Kim et al. (2025) and SSF Khoche et al. (2025) on a mixture of nuScenes, AV2, and Waymo. Importantly, we do not re-weight dataset frequencies or apply dataset-specific augmentations. Following prior work, we apply height augmentation to account for sensor placement variations and introduce random LiDAR ray

Table 1: **Analysis on Cross-Domain Generalization.** We train Flow4D (Kim et al., 2025) on AV2, Waymo and nuScenes and evaluate each dataset-specific model across datasets. We find that Flow4D (AV2) achieve similar Dynamic Mean EPE on AV2 and Waymo (8.55 vs. 8.31), suggesting that it *already* generalizes across datasets. Similarly, Flow4D (Waymo) nearly matches Flow4D (AV2)'s overall performance on AV2 (0.0890 vs. 0.0855), and outperforms it on fast moving objects (9.27 vs. 10.33). Flow4D (nuScenes) performs considerably worse on out-of-distribution datasets due to the sparsity of its LiDAR and limited number of training examples.

| Train Data | Test Data | FD (cm) | Dyn. Mean | [0, 0.5) | [0.5, 1.0) | [1.0, 2.0) | [2.0, ∞) |
|---|---|---|---|---|---|---|---|
| AV2 | AV2 | 8.55 | 0.1918 | 0.0196 | 0.1190 | 0.1141 | 0.1033 |
| | Waymo | 8.31 | 0.2356 | 0.0161 | 0.0772 | 0.0718 | 0.3498 |
| | nuScenes | 20.40 | 0.4629 | 0.0174 | 0.2492 | 0.3335 | 0.6101 |
| Waymo | AV2 | 8.90 | 0.1985 | 0.0210 | 0.1190 | 0.1249 | 0.0927 |
| | Waymo | 4.58 | 0.2145 | 0.0068 | 0.0641 | 0.0508 | 0.1301 |
| | nuScenes | 16.04 | 0.4084 | 0.0116 | 0.2166 | 0.3189 | 0.5401 |
| nuScenes | AV2 | 16.08 | 0.3565 | 0.0386 | 0.2388 | 0.2724 | 0.3954 |
| | Waymo | 17.24 | 0.3642 | 0.0223 | 0.2219 | 0.2092 | 0.5982 |
| | nuScenes | 7.97 | 0.2304 | 0.0127 | 0.0998 | 0.1086 | 0.3703 |

dropping to simulate sparser sensors. These simple augmentations significantly enhance UniFlows generalization to out-of-domain data (cf. Table 7).

**Test-Time Optimization.** Despite the UniFlow's impressive zero-shot performance, we note that it underestimates flow vectors for fast moving objects in TruckScenes, likely because nuScenes, Argoverse, and Waymo primarily collect data in urban city centers, while TruckScenes collects data on highways. Therefore, we repurpose, Neural Scene Flow Prior (NSFP) Li et al. (2021) to learn residual flow vectors to mitigate UniFlow's dataset-specific biases. NSFP uses the inductive bias of the smooth, restricted learnable function class of two ReLU MLP coordinate networks (8 hidden layers of 128 neurons); $\theta$ to estimate forward flow and $\theta'$ to estimate backwards flow, minimizing

$$\text{TruncatedChamfer}(P_t + \theta\left(P_t\right), P_{t+1}) + \left\|P_t + \theta\left(P_t\right) + \theta'\left(P_t + \theta\left(P_t\right)\right) - P_t\right\|_2 \ ,$$

where TruncatedChamfer is defined as the standard $L_2$ Chamfer distance, with per-point distances above 2 meters set to zero in order to reduce the influence of outliers. NSFP is optimized for at most 5000 steps with early stopping. We modify the loss function to predict residual flows by minimizing

$$\text{TruncatedChamfer}(P_t + UniFlow(.) + \theta\left(P_t\right), P_{t+1}) +$$
$$\left\|P_t + UniFlow(.) + \theta\left(P_t\right) + \theta'\left(P_t + UniFlow(.) + \theta\left(P_t\right)\right) - P_t - UniFlow(.)\right\|_2$$

We find that this residual network's predictions are unreliable for static objects, so we only advect the residual to moving points according to UniFlow's original predictions. We threshold all points moving below 10.0 m/s as static. Notably, initializing NSFP with UniFlow's predictions speeds up convergence rates by 10%. We report results with the extra Test-Time Optimization step in the ablation studies.

## 4 EXPERIMENTS

**Datasets** We train and evaluate our model on four large-scale autonomous driving datasets, selected to cover diverse sensor configurations, ego-vehicle platforms, and driving scenarios critical for studying cross-domain generalization (cf. Table 4). Argoverse 2 (AV2) Wilson et al. (2021) and nuScenes Caesar et al. (2020) provide data from sedans operating in dense urban environments, collected with two 32-beam LiDARs and a single 32-beam LiDAR, respectively. The Waymo Open Dataset (WOD) Sun et al. (2020) offers denser point clouds from a custom LiDAR, spanning a mix of urban and suburban scenes with more varied traffic dynamics. To test zero-shot generalization, we use TruckScenes Fent et al. (2024), captured from a large truck equipped with two 64-beam long-range LiDARs, primarily in high-speed highway settings. While AV2 and WOD provide standard 10 Hz LiDAR data, nuScenes and TruckScenes require special treatment due to sparse 2 Hz annotations that are misaligned with their higher-frequency sensor data. To construct a consistent 10 Hz benchmark across all datasets, we generate ground-truth flow from official velocity labels at 10 Hz and apply LineFit Himmelsbach et al. (2010) for ground removal. These four datasets allow us to systematic analyze the impact of point density (32-beam vs. 64-beam), ego-vehicle perspective (car vs. truck), and velocity (urban vs. highway) on cross-domain generalization.

**Metrics** To evaluate cross-domain generalization, particularly across diverse velocity distributions, our primary metric is the *Dynamic Bucket-Normalized EPE* from Khatri et al. (2024). This protocol is specifically designed to be both class-aware and speed-normalized. By normalizing the error by

Table 2: **Dataset Statistics.** We summarize the four datasets used for training and evaluation below. We include both the (# Total Frames / # Annotated Frames) for nuScenes and TruckScenes.

| Dataset | Split | Scenes | Frames | LiDAR Setup | Ego-Vehicle | Scenario |
|---------|-------|--------|--------|-------------|-------------|----------|
| Argoverse 2 | Train | 700 | 110,071 | 2 x 32-beam | Car | Urban City |
| | Val | 150 | 23,547 | | | |
| Waymo | Train | 798 | 155,000 | 1 x 64-beam | Car | Mixed |
| | Val | 202 | 40,000 | | | |
| nuScenes | Train | 700 | 137,575 / 27,392 | 1 x 32-beam | Car | Urban City |
| | Val | 150 | 29,126 / 5,798 | | | |
| TruckScenes | Train | 524 | 101,902 / 20,380 | 2 x 64-beam | Truck | Highway |
| | Val | 75 | 14,625 / 2,925 | | | |

an object's speed, it measures the fraction of motion not described, enabling a fair and insightful comparison between slow-moving pedestrians and fast-moving vehicles. This detailed, per-speed-bucket analysis is crucial for validating our core insight that velocity distribution is a primary factor in scene flow generalization. For completeness and comparison with prior work, we also report the standard Three-way and dynamic EPE Chodosh et al. (2024).

Table 3: **Argoverse 2 Test Set.** We compare SSF (UniFlow) and Flow4D (UniFlow) with recent methods on the Argoverse 2 test set. Notably, Flow4D (UniFlow) outperforms Flow4D by 11.41%, while SSF (UniFlow) outperforms SSF by 17.89% Dynamic Mean EPE. Flow4D (UniFlow) achieves parity with EulerFlow on Dynamic Mean EPE (0.132 vs. 0.130) at a fraction of the runtime.

| Method | Three-way EPE (cm) | | | | Dynamic Bucket-Normalized | | | | |
|---|---|---|---|---|---|---|---|---|---|
| | Mean ↓ | FD ↓ | FS ↓ | BS ↓ | Dyn. Mean ↓ | Car ↓ | Other ↓ | Pedestr. ↓ | VRU ↓ |
| *Unsupervised* | | | | | | | | | |
| NSFP | 6.06 | 11.58 | 3.16 | 3.44 | 0.422 | 0.251 | 0.331 | 0.722 | 0.383 |
| FastNSF | 11.18 | 16.34 | 8.14 | 9.07 | 0.383 | 0.296 | 0.413 | 0.500 | 0.322 |
| SeFlow | 4.86 | 12.14 | 1.84 | 0.60 | 0.309 | 0.214 | 0.291 | 0.464 | 0.265 |
| ICP Flow | 6.50 | 13.69 | 3.32 | 2.50 | 0.331 | 0.195 | 0.331 | 0.435 | 0.363 |
| EulerFlow | 4.23 | 4.98 | 2.45 | 5.25 | **0.130** | 0.093 | 0.141 | 0.195 | 0.093 |
| *Supervised* | | | | | | | | | |
| TrackFlow | 4.73 | 10.30 | 3.65 | 0.24 | 0.269 | 0.182 | 0.305 | 0.358 | 0.230 |
| DeFlow | 3.43 | 7.32 | 2.51 | 0.46 | 0.276 | 0.113 | 0.228 | 0.496 | 0.266 |
| SSF | 2.89 | 5.93 | 1.82 | 0.91 | 0.190 | 0.110 | 0.175 | 0.295 | 0.177 |
| Flow4D | 2.45 | 4.98 | 1.70 | 0.67 | 0.149 | 0.092 | 0.139 | 0.237 | 0.130 |
| SSF (UniFlow, Ours) | 2.23 | 4.89 | 1.31 | 0.51 | 0.156 | 0.102 | 0.147 | 0.241 | 0.134 |
| Flow4D (UniFlow, Ours) | **2.07** | **4.50** | **1.31** | **0.40** | 0.132 | **0.072** | **0.131** | **0.218** | **0.107** |

Table 4: **Waymo Validation Set.** We compare SSF (UniFlow) and Flow4D (UniFlow) with recent methods on Waymo val. Notably, supervised methods achieve significantly lower EPE than unsupervised methods, with Flow4D (UniFlow) beating SeFlow by 74.57% on Foreground Dynamic.

| Method | Three-way EPE (cm) | | | |
|---|---|---|---|---|
| | Mean ↓ | FD ↓ | FS ↓ | BS ↓ |
| *Unsupervised* | | | | |
| NSFP | 10.05 | 17.12 | 10.81 | 2.21 |
| SeFlow | 5.98 | 15.06 | 1.81 | 1.06 |
| ZeroFlow | 8.52 | 21.62 | 1.53 | 2.41 |
| *Supervised* | | | | |
| FastFlow3D | 7.84 | 19.54 | 2.46 | 1.52 |
| DeFlow | 4.46 | 9.80 | 2.59 | 0.98 |
| SSF | 2.19 | 5.62 | 0.74 | 0.23 |
| Flow4D | 1.82 | 4.58 | 0.61 | 0.28 |
| SSF (UniFlow, Ours) | 1.85 | 4.74 | 0.58 | 0.22 |
| Flow4D (UniFlow, Ours) | **1.60** | **3.83** | **0.68** | **0.30** |

Table 5: **nuScenes Validation Set.** We compare SSF (UniFlow) and Flow4D (UniFlow) with recent methods on the nuScenes val set. Interestingly, SSF (UniFlow) outperforms Flow4D (UniFlow) by 26.53% Dynamic Mean EPE, suggesting that SSF's architecture more effectively learns to estimate flow from sparse point clouds.

| Method | Three-way EPE (cm) | | | | Dynamic Bucket-Normalized | | | | |
|---|---|---|---|---|---|---|---|---|---|
| | Mean ↓ | FD ↓ | FS ↓ | BS ↓ | Dyn. Mean ↓ | Car ↓ | Other ↓ | Pedestr. ↓ | VRU ↓ |
| *Unsupervised* | | | | | | | | | |
| NSFP | 10.79 | 20.26 | 4.88 | 7.23 | 0.602 | 0.463 | 0.456 | 0.829 | 0.662 |
| SeFlow | 8.19 | 16.15 | 3.97 | 4.45 | 0.554 | 0.396 | 0.635 | 0.726 | 0.419 |
| *Supervised* | | | | | | | | | |
| DeFlow | 3.98 | 6.99 | 3.45 | 1.50 | 0.314 | 0.163 | 0.286 | 0.533 | 0.275 |
| SSF | 3.00 | 6.55 | 2.04 | 0.41 | 0.220 | 0.142 | 0.197 | 0.398 | 0.144 |
| Flow4D | 3.46 | 7.97 | 1.85 | 0.55 | 0.230 | 0.160 | 0.241 | 0.345 | 0.176 |
| SSF (UniFlow, Ours) | **1.97** | **4.33** | **1.38** | **0.20** | **0.144** | **0.081** | **0.131** | **0.267** | **0.097** |
| Flow4D (UniFlow, Ours) | 3.01 | 7.28 | 1.47 | 0.28 | 0.196 | 0.137 | 0.219 | 0.272 | 0.157 |

**Comparison to State-of-the-Art Methods.** In order to demonstrate the power of the Uni-Flow framework we choose two popular state of the art supervised scene flow methods, SSF and Flow4D and train them using UniFlow's unified datasets and augmentation regime. We evaluate SSF (UniFlow) and Flow4D (UniFlow) against recent approaches on multiple benchmarks (Tables 3-5). On the Argoverse 2 test set, Flow4D (UniFlow) improves over Flow4D by 11.41% and SSF (UniFlow) improves over SSF by 17.89% in Dynamic Mean EPE. Moreover, Flow4D (Uni-Flow) achieves near-parity with EulerFlow (a previous SOTA method) while running at a fraction

Table 6: **TruckScenes Validation Set.** We compare Flow4D (UniFlow) with recent methods on the TruckScenes validation set. According to Three-Way EPE, Flow4D achieves the best performance due to low Foreground Dynamic error. However, Dynamic Bucket Normalized EPE highlights that Flow4D (UniFlow) outperforms it across all moving object categories, most notably on pedestrians and VRUs.

| Method | Three-way EPE (cm) | | | | Dynamic Bucket-Normalized | | | | |
|---|---|---|---|---|---|---|---|---|---|
| | Mean ↓ | FD ↓ | FS ↓ | BS ↓ | Dyn. Mean ↓ | Car ↓ | Other ↓ | Pedestr. ↓ | VRU ↓ |
| *Unsupervised* | | | | | | | | | |
| NSFP | 45.63 | 120.45 | 2.07 | 14.38 | 0.658 | 0.303 | 0.350 | 1.221 | 0.758 |
| FastNSF | 30.72 | 59.44 | 3.35 | 29.38 | 0.588 | 0.218 | 0.376 | 1.124 | 0.635 |
| ICP Flow | 58.82 | 169.91 | 1.43 | 5.12 | 0.472 | 0.302 | 0.614 | 0.596 | 0.376 |
| *Supervised* | | | | | | | | | |
| DeFlow | 7.30 | 16.47 | 1.67 | 3.77 | 0.570 | 0.180 | 0.410 | 0.970 | 0.730 |
| Flow4D | 16.14 | 44.87 | 1.71 | 1.85 | 0.456 | 0.176 | 0.351 | 0.885 | 0.413 |
| SSF (UniFlow, Ours) | 35.23 | 103.72 | 1.69 | 0.27 | 0.435 | 0.149 | 0.455 | 0.669 | 0.466 |
| Flow4D (UniFlow, Ours) | 23.59 | 68.41 | 1.78 | 0.57 | 0.281 | 0.088 | 0.277 | 0.530 | 0.230 |

Table 7: **Ablation on TruckScenes.** We ablate the impact of multi-dataset training, data augmentation and test-time optimization on zero-shot TruckScenes performance. First, we find that both Flow4D and SSF benefit from training on multiple datasets. Interestingly, despite training on a 3X larger dataset, data augmentation still provides significant benefits, suggesting that further scaling may improve performance further. We find that test-time optimization slightly improves Foreground Dynamic EPE and Mean Dynamic EPE for fast moving objects.

| Method | FD | Dynamic EPE | [0, 0.5) | [0.5, 1.0) | [1.0, 2.0) | [2.0, ∞) |
|---|---|---|---|---|---|---|
| Flow4D | 116.01 | 0.336 | 0.071 | 0.129 | 0.215 | 0.586 |
| + Unified Dataset | 93.76 | 0.310 | 0.040 | 0.116 | 0.170 | 0.594 |
| + Augmentation | 68.32 | 0.301 | 0.047 | 0.111 | 0.141 | 0.407 |
| + XL Backbone | 68.41 | 0.281 | 0.033 | 0.097 | 0.119 | 0.389 |
| + Optimization | 65.57 | 0.284 | 0.034 | 0.099 | 0.130 | 0.368 |
| SSF | 170.65 | 0.737 | 0.049 | 0.555 | 0.740 | 0.971 |
| + Unified Dataset | 133.04 | 0.577 | 0.042 | 0.257 | 0.514 | 0.827 |
| + Augmentation | 103.72 | 0.435 | 0.036 | 0.149 | 0.285 | 0.637 |
| + Optimization | 89.77 | 0.460 | 0.036 | 0.177 | 0.332 | 0.577 |

of the cost. Similar improvements can be seen between the stock Flow4D and SSF networks and their UniFlowcounterparts on the Waymo validation set in Table 4. In Table 5 we similarly evaluate on the NuScenes validation set. Interestingly the SSF model architecture when paired with the UniFlow augmentations and multi-dataset training outperforms the Flow4D architecture under the same regime, suggesting that the SSF architecture is naturally more suited to the sparser point clouds present in the NuScenes dataset.

**Zero-Shot TruckScenes Performance** In Table 6 we demonstrate UniFlow's strong zero-shot generalization capabilities. All baseline methods were trained on TruckScenes and evaluated on the same. However for UniFlow we use the exact same checkpoint used for the previous AV2, Waymo, and nuScenes experiments which has notably never seen any TruckScenes data during training. Despite this, Flow4D (UniFlow) significantly outperforms Flow4D with a dataset specific checkpoint. This is despite the significant domain gap between the datasets used for training UniFlow (which are all urban car datasets) and TruckScenes which is a highway trucking dataset. Table 6 also demonstrates the importance of the speed normalized and bucketed EPE metric from Khatri et al. (2024), as Threeway-EPE results don't adequately measure the significant improvements that Flow4D (UniFlow) makes on pedestrian and VRU performance.

**Ablation on Frame-Rate** We evaluate the generalization of Flow4D (UniFlow) and SSF (UniFlow) across different frame rates in Table 8. Slower frame rates approximate a slower-moving ego-vehicle, while faster frame rates approximate a faster one. Although all models are trained using 10 Hz annotations, multi-dataset training consistently improves performance, especially at slower frame rates. Interestingly, higher frame rates have a smaller negative effect on performance.

**Ablation on TruckScenes.** We ablate the effects of multi-dataset training, data augmentation, and test-time optimization on zero-shot TruckScenes performance. Both Flow4D and SSF show clear gains from multi-dataset training. Notably, even with a dataset three times larger, data augmentation continues to yield substantial improvements, indicating that further scaling could drive additional performance gains. In addition we add Test-Time optimization as an extra step for the UniFlow methods. We notice that despite better foreground dynamic Threeway-EPE performance, and better performance on fast moving objects in Mean Dyanmic EPE, there are slight regressions

Table 8: **Ablation on Frame-Rate.** We evaluate Flow4D (UniFlow) and SSF (UniFlow)'s generalization to different frame rates. Note that slower frame-rates effectively mimic a slower ego-vehicle, while faster-frame rates mimic a faster ego-vehicle. Although we train all models on 10 Hz annotations, multi-dataset training yields better results, particularly at slower frame rates. Interestingly, faster frame rates do not negatively impact performance nearly as much as slower frame-rates.

| Method | Three-way EPE (cm) | | | | Dynamic Bucket-Normalized | | | | |
|---|---|---|---|---|---|---|---|---|---|
| | Mean ↓ | FD ↓ | FS ↓ | BS ↓ | Dyn. Mean ↓ | Car ↓ | Other ↓ | Pedestr. ↓ | VRU ↓ |
| *2 Hz* | | | | | | | | | |
| Flow4D | 30.79 | 88.10 | 3.26 | 1.02 | 0.495 | 0.364 | 0.494 | 0.613 | 0.509 |
| SSF | 31.16 | 89.82 | 3.07 | 0.59 | 0.520 | 0.363 | 0.498 | 0.684 | 0.535 |
| Flow4D (UniFlow, Ours) | **27.52** | **79.54** | **2.36** | 0.65 | **0.444** | 0.297 | **0.415** | **0.587** | **0.476** |
| SSF (UniFlow, Ours) | 28.67 | 83.11 | 2.50 | **0.39** | 0.503 | **0.296** | 0.432 | 0.671 | 0.611 |
| *5 Hz* | | | | | | | | | |
| Flow4D | 9.58 | 25.64 | 2.45 | 0.65 | 0.323 | 0.215 | 0.312 | 0.432 | 0.333 |
| SSF | 8.74 | 23.02 | 2.74 | 0.47 | 0.311 | 0.179 | 0.286 | 0.489 | 0.291 |
| Flow4D (UniFlow, Ours) | 8.02 | 21.78 | **1.92** | 0.37 | 0.290 | 0.190 | 0.265 | **0.405** | **0.302** |
| SSF (UniFlow, Ours) | **6.75** | **17.96** | 2.06 | **0.22** | **0.285** | **0.135** | **0.218** | 0.454 | 0.333 |
| *10 Hz (Standard Frame Rate)* | | | | | | | | | |
| Flow4D | 3.46 | 7.97 | 1.85 | 0.55 | 0.230 | 0.160 | 0.241 | **0.241** | 0.176 |
| SSF | 3.00 | 6.55 | 2.04 | 0.41 | 0.220 | 0.142 | 0.197 | 0.398 | 0.144 |
| Flow4D (UniFlow, Ours) | 3.01 | 7.28 | 1.47 | 0.28 | 0.196 | 0.137 | 0.219 | 0.272 | 0.157 |
| SSF (UniFlow, Ours) | **1.97** | **4.33** | **1.38** | **0.20** | **0.144** | **0.081** | **0.131** | 0.267 | **0.097** |
| *20 Hz* | | | | | | | | | |
| Flow4D | 2.55 | 5.79 | 1.35 | 0.50 | 0.272 | 0.196 | 0.315 | 0.377 | 0.200 |
| SSF | 2.94 | 6.72 | 1.73 | 0.38 | 0.316 | 0.177 | 0.303 | 0.533 | 0.250 |
| Flow4D (UniFlow, Ours) | **1.95** | **4.65** | **1.00** | **0.21** | **0.230** | 0.171 | 0.269 | **0.315** | **0.163** |
| SSF (UniFlow, Ours) | 2.13 | 5.16 | 1.11 | 0.12 | 0.246 | **0.127** | 0.275 | 0.395 | 0.189 |

for slower moving objects which yield worse overall Dynamic EPE. While we do not include our Test-Time Optimization technique as a core component of UniFlow we believe it to be a useful avenue for future exploration.

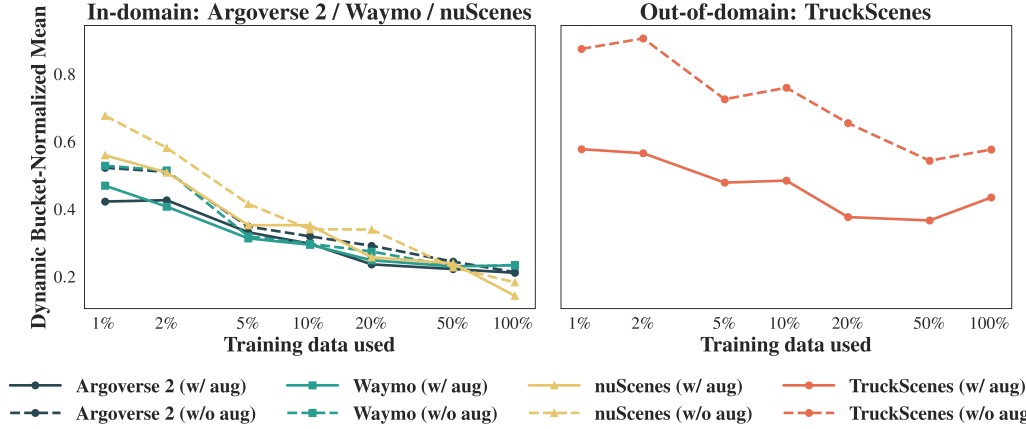

Figure 4: **Scaling Laws.** We evaluate the performance of Flow4D (UniFlow) with different amounts of training data both in-distribution (on AV2, nuScenes, and Waymo), and out-of-distribution (on TruckScenes). Unsurprisingly, increasing data reduces Dynamic Mean EPE. However, we find that data augmentation is significantly more important for out-of-distribution performance, and has minimal impact on in-distribution performance.

**Scaling Laws.** We evaluate Flow4D (UniFlow) with varying amounts of training data on both in-distribution benchmarks (AV2, nuScenes, and Waymo) and an out-of-distribution benchmark (TruckScenes) in Fig 4. As expected, larger training sets reduce Dynamic Mean EPE. However, data augmentation proves far more critical for out-of-distribution performance, while its effect on in-distribution performance remains minimal.

**Analysis of Failure Cases.** The first failure case (top row of Fig. 5) illustrates artifacts in the predicted scene flow. It comes from a truck yard with many parked trailers and heavy occlusions. This is quite different from the urban environments in the training data, and UniFlow ends up predicting motion that does not actually exist. The second failure case (bottom row of Fig. 5) is from a rainy

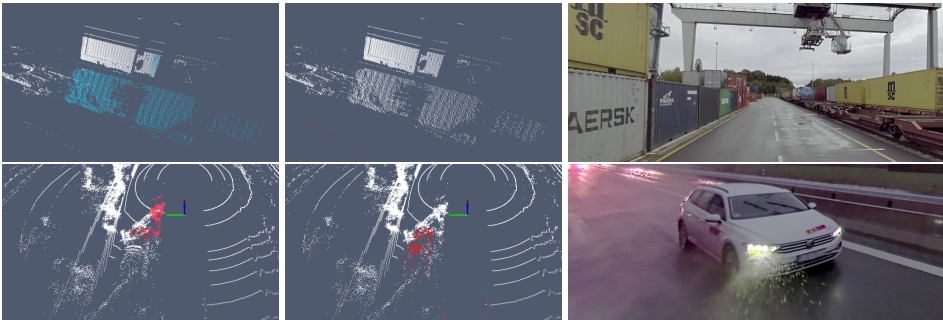

Figure 5: **Visuals of Failure Cases.** Each row shows one failure example with (a) UniFlow prediction, (b) ground truth, and (c) RGB camera frame. Top row: a truck yard with many parked trailers and occlusions, leading to artifacts in UniFlows prediction. Bottom row: a rainy scene with heavy LiDAR noise, where UniFlowfails to capture motion and predicts artifacts.

scene, where the LiDAR data is highly noisy. In this case, UniFlowfails to capture most of the true motion and instead predicts artifacts on the noise.

**Limitations and Future Work** In this paper we investigate the domain gap between various LiDAR scene flow datasets and demonstrate that strong generalization can arise from multi-datsaet training despite differences in the sensing suite due to the low-level signal provided by the task of scene flow. However we are limited to quantitative evaluations on autonomous vehicle datasets due to a lack of scene flow benchmarks and baselines in other domains. Future work should investigate non-AV domains and continue our efforts to combine feed-forward methods with Test-Time Optimization techniques.

## 5 CONCLUSION

In this paper, we introduce UniFlow, a frustratingly simple approach that re-trains off-the-shelf Li-DAR scene flow models with diverse data from multiple datasets. Although prior work in LiDAR-based 3D object detection and segmentation don't seem to benefit from multi-dataset training, we posit that our proposed method works well because dataset-specific LiDAR-based scene flow models *already* achieve strong cross-domain generalization. Notably, our model establishes a new state-of-the-art Waymo and nuScenes, improving over prior work by 16.4% and 34.5% respectively. Although UniFlowis primarily trained on urban city driving scenarios, we demonstrate that it generalizes surprisingly well to highway truck driving *without any dataset-specific fine-tuning*. Our extensive analysis shows that such improvements are model-agnostic, suggesting that future scene flow methods should adopt a multi-dataset training strategy.

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
