## A IMPLEMENTATION DETAILS

**Model.** We use SSF exactly as released and report its size as 38.8 million parameters. For Flow4D we report the original model with 4.60 million parameters and an enlarged variant named Flow4D-XL. Flow4D-XL keeps the published architecture unchanged in depth, kernel sizes, strides, temporal context of 5 frames, and voxel grid of 51251232. The only modification is width scaling by a constant factor of three applied to every convolutional layer in the backbone so that all input and output channel counts are tripled, and the hidden channels of the point-wise prediction head are also tripled, while the final flow layer retains three output channels. No other hyperparameters or activations are changed. This change makes the model size to 41.4 million parameters.

Table 9: **Details of 4D voxel network of Flow4D-XL.**

| Stage | STDB-P (Filters) | Pool & Up (Kernel, Stride) | Output shape $[W \times L \times H \times T \times ch]$ |
|---|---|---|---|
| Input | – | – | $512 \times 512 \times 32 \times 5 \times \mathbf{48}$ |
| 1 | (48, 96)
(96, 96) | Pool (2,2,2,1) | $256 \times 256 \times 16 \times 5 \times \mathbf{96}$ |
| 2 | (96, 192)
(192, 192) | Pool (2,2,2,1) | $128 \times 128 \times 8 \times 5 \times \mathbf{192}$ |
| 3 | (192, 192)
(192, 192) | Pool (2,2,2,1) | $64 \times 64 \times 4 \times 5 \times \mathbf{192}$ |
| 4 | (192, 192)
(192, 192) | Pool (2,2,1,1) | $32 \times 32 \times 4 \times 5 \times \mathbf{192}$ |
| 5 | (192, 192)
(192, 192) | Up (2,2,1,1) | $32 \times 32 \times 4 \times 5 \times \mathbf{192}$ |
| 6 | (192, 192)
(192, 192) | Up (2,2,2,1) | $64 \times 64 \times 4 \times 5 \times \mathbf{192}$ |
| 7 | (192, 192)
(192, 192) | Up (2,2,2,1) | $128 \times 128 \times 8 \times 5 \times \mathbf{192}$ |
| 8 | (192, 192)
(192, 192) | Up (2,2,2,1) | $256 \times 256 \times 16 \times 5 \times \mathbf{192}$ |
| 9 | (96, 48) | – | $512 \times 512 \times 32 \times 5 \times \mathbf{48}$ |

**Augmentations.** We use two dataset-agnostic augmentations. First, height jitter identical to DeltaFlow: with probability 0.8, we translate the point cloud along the sensor z-axis by a uniform offset in [0.5, 2.0] m. Second, beam dropout to reduce cross-dataset density mismatch: with probability 0.35, we remove every other LiDAR beam for all lidar scans in the subsequence, approximately halving the active beams to emulate a 32-beam sensor.