# OpenReview forum: "UniFlow: Zero-Shot LiDAR Scene Flow via Cross-Domain Generalization"
_ICLR.cc/2026/Conference — ICLR 2026 Conference Withdrawn Submission_

### Official Review · Reviewer_xV5t · 2025-10-27

**Soundness:** 3
**Presentation:** 3
**Contribution:** 2
**Rating:** 2
**Confidence:** 5

**Summary:**

This paper proposed a multi-dataset training strategy for better performance and generalization ability in scene flow models. Although the multi-dataset training can improve the performance, I have concerns about the zero-shot ability and efficiency.

**Strengths:**

1.The overall motivation seems great. Some test-time adaptation techniques show potential in zero-shot generalization, thus it makes sense transferring them into scene flow topic.

2.Experiments demonstrate the state-of-the-art performance when adding the proposed multi-dataset training.

3.Authors showcase some failure cases and illustrate the reason.

**Weaknesses:**

1.The main concern is limited novelty, especially for the technique contribution. Authors only leverage more datasets for training, thus providing a better accuracy. However, there seem to be no technique contributions to this community. Plus, a better performance by using more data is somehow intuitive.

2.As for unifying different frame rates in sensors, will the linear interpolation method incur inaccuracy? Some recent works, e.g., LiDAR4D (CVPR24), NeuralPCI (CVPR23) indicated that the 4D motion in real worlds are non-linear and using a linear model will introduce large errors. Plus, in the ablation study, it seems that the different frame setting can significantly affect the performance. Does this prove the model is sensitive to frame rating?

3.In the test-time optimization section, since NSFP has been criticized by large time overhead (FastNSF, ICCV23), could this model introduce large computational overhead?

4.In the forward and backward loss, why authors first add Pt and then minus it? How about directly using the subtract between forward and backward flows?

5.‘so we only advect the residual to moving points according to UniFlow’s original predictions’. Here, there lack some descriptions about what kind of classification methods are used, i.e., how to distinguish dynamic and static points? Also, will the threshold be too high to capture some subtle motions? Since 10m/s is basically the speed of vehicles. But some other agents like pedestrians, cyclists, slowly-moving cars also have motions. More explanations and ablation studies about this threshold are expected.

6.I doubt the setting in Table 6. Authors compare other methods trained on TrackScene and the multi-dataset training (UniFlow) trained on other datasets like Waymo. But the overall performance is worse than methods trained on TrackScene. So what conclusion can this experiment lead to? I doubt authors should train other methods on out-of-domain datasets like waymo and maybe the performance is bad. Then, adding multi-dataset training can improve the performance. This can demonstrate the advantages of the proposed method.

7.Authors should give an evaluation about efficiency. In AD scenes, real-time application is crucial.

**Questions:**

Please see the weakness parts. I think the current version cannot backup the zero-shot claim. In contrast, Zero-msf (CVPR25) can predict scene flow from any online videos.

---

### Official Review · Reviewer_vim5 · 2025-11-01

**Soundness:** 1
**Presentation:** 2
**Contribution:** 1
**Rating:** 2
**Confidence:** 4

**Summary:**

This paper argues that the lidar scene flow estimation can be trained together on multiple datasets and achieves better generalizable performance across different datasets. The authors have proposed to re-train the previous SOTA methods across different lidar scene flow datasets, such as Argoverse2, Waymo, nuScenes, and TruckScenes, improving their performance on “out-of-distribution” testing. In addition, the authors also propose to use NSFP to finetune the scene flow estimation when testing on a specific dataset, further mitigating the residual errors. Using this cross-domain dataset, re-training and test-time fine-tuning, the proposed technique has achieved good performance across different lidar scene flow datasets.

**Strengths:**

- The re-training and fine-tuning technique used in the paper is good for cross-domain lidar scene flow estimation.
- Focusing on data distribution of lidar scene flow estimation and how to mitigate the cross-domain training is interesting and has a potential benefit for the autonomous driving research.

**Weaknesses:**

- The authors have stated that one of the contributions of the paper is that “dataset-specific scene flow models already achieve strong performance across datasets, challenging conventional wisdom about LiDAR-based cross-domain generalization”. However, when we talk about cross-domain generalization in scene flow, we often argue about cross-domain distribution shifts, especially for synthetic datasets and real-world datasets. When focusing on lidar scene flow cross-domain generalization, we discuss the sensor difference, the long-tailed distribution in motions, etc. As also shown in the paper, Argoverse2 and Waymo datasets share a similar data distribution with the same lidar sampling rate, except for the fast-moving objects. And the estimation of previous data-specific training methods on both datasets shares a similar performance. However, nuScenes has a much lower sampling rate, while TruckScenes has more fast-moving objects. The estimation of previous data-specific training methods cannot generalize well. This does not “challenge conventional wisdom about LiDAR-based cross-domain generalization”. In addition, to make this contribution valid, the authors should provide more detailed analysis and discussions.
- The paper has shown different training settings with different methods on different datasets. However, to further explain the validation of the quantitative results, there lacks reasonable analysis, such as theoretical analysis, data distribution analysis, training analysis, etc. Otherwise, this paper only shows the results/performance, but lacks reasoning and intuitive discussions, making it more like a technical report without a detailed analysis.
- When looking into the failure cases, it seems that the proposed re-training on multiple datasets does not solve the existing issues in the current lidar scene flow methods.
- The performance on TruckScenes is not “universally good” for improving the original methods.

Minor suggestions:
- The reference link is not working in the paper, and there might be some omitted references.
- Figure 1 could add a zoomed-in figure to show the moving objects.
- Table 6 omitted the performance for the original SSF method.

**Questions:**

Please see the above comments for details. I encourage the authors to look into the grounded analysis of this cross-domain generalization problem for the lidar scene flow and revise the paper.

---

### Official Review · Reviewer_YXRw · 2025-11-01

**Soundness:** 3
**Presentation:** 3
**Contribution:** 2
**Rating:** 4
**Confidence:** 4

**Summary:**

This paper challenges the conventional wisdom in LiDAR perception. Previously, researchers think that multi-dataset training is harmful due to domain gaps in sensor setup and environment. The authors argue that scene flow is less sensitive to these domain shifts compared to high-level tasks like detection and segmentation. They propose UniFlow, a method that involves training scene flow models (Flow4D, SSF) on a mixture of multiple autonomous driving datasets (Argoverse 2, Waymo, nuScenes). The key finding is that this multi-dataset training not only improves in-domain performance but also leads to strong zero-shot generalization (e.g., TruckScenes).

**Strengths:**

- The key finding that multi-dataset training is beneficial for LiDAR scene flow is interesting, which contradicts  common beliefs in the field. This insight could significantly influence how the community approaches data scaling and generalization for 3D motion tasks.
- The experimental results are impressive to me. The UniFlow achieves new state-of-the-art or competitive performance on several major benchmarks (Waymo, nuScenes, Argoverse 2) and demonstrates zero-shot capabilities on TruckScenes.
- The claimed pipeline is easy to follow.

**Weaknesses:**

- The most significant concern to me is the lack of an investigation into why scene flow is less sensitive to domain gaps (e.g., scenes, LiDAR config). The authors explained that  this is because scene flow is a low-level task. A key experiment would be to take a single dataset (e.g., Waymo) and artificially degrade the point clouds (e.g., from 64-beam to 32-beam) to create a controlled domain gap. Training and evaluating on this controlled setup would isolate the effect of point density and provide much stronger evidence for their hypothesis.
- Using mixtures of training datasets is a common practice for  low-level domains such as  depth estimation. For AD, many works train their monocular depth estimator in KITTI, nuScenes, DAAD, and etc.
- The performance gains might be primarily attributable to the increased volume and diversity of data. It is plausible that scene flow is a data-hungry task where any additional data is beneficial, and that the harmful effects of domain gap would become apparent if even more heterogeneous datasets were added.
- The work is compared to dataset-specific models. A more robust evaluation would include comparisons with modern domain adaptation.

**Questions:**

- Could you design an experiment to disentangle the benefit of simply having more data from the benefit of overcoming a domain gap? For instance, does adding more data from a single dataset provide similar gains to adding a different dataset?

- Have you considered or experimented with any domain adaptation methods? How does your multi-dataset training approach compare?

- Given the computational cost of NSFP, can you discuss the real-world applicability of your full pipeline?

I give a minor rejection at this stage because I have some concerns about data augmentation. I hope the authors can discuss these questions.

---

### Official Review · Reviewer_GBR1 · 2025-11-01

**Soundness:** 3
**Presentation:** 3
**Contribution:** 2
**Rating:** 6
**Confidence:** 3

**Summary:**

The paper proposes UniFlow, a “frustratingly simple” recipe for zero-shot LiDAR scene flow via cross-dataset training. The authors unify multiple AV datasets (Argoverse 2, Waymo, nuScenes; TruckScenes for zero-shot) by standardizing frame rate and annotations, then retrain existing feedforward SOTA models (e.g., Flow4D, SSF) jointly across datasets without dataset-specific reweighting or bespoke augmentations. UniFlow achieves SOTA on nuScenes and Waymo and strong performance on AV2, while also exhibiting remarkable zero-shot generalization on the out-of-domain TruckScenes benchmark. The paper further introduces a test-time optimization (TTO) add-on—residual refinement using NSFP—trading extra latency for small gains on fast-moving objects.

**Strengths:**

1. UniFlow demonstrates consistent improvements across AV2/nuScenes/Waymo when training jointly, overturning the common belief that LiDAR cross-dataset training hurts performance for perception tasks.

2. UniFlow shows strong zero-shot ability. A single checkpoint generalizes to TruckScenes (highway, truck platform) with big margins under speed-normalized metrics—despite never seeing that dataset during training.

3. The approach reuses prior SOTA architectures (Flow4D, SSF) and shows that cross-data training plus lightweight augmentations already confer large performance benefits—highlighting the scaling value of diverse data over bespoke architecture.

4. Thorough comparisons and ablations across datasets/frame rates, plus a failure-case analysis and scaling-law study.

5. Presentation is clear, with intuitive figures and tables.

**Weaknesses:**

1. Test-time optimization overhead. The NSFP residual-refinement step adds latency and compute; it yields modest improvements on fast movers but may undermine real-time applicability.

2. The paper argues that low-level tasks such as scene flow are less sensitive to sensor differences and benefit more from cross-dataset training. However, the underlying reasons for their strong performance—especially compared to label-intensive tasks like detection and segmentation—call for more causal analysis beyond the compelling velocity-distribution evidence provided.
On the other hand, it is well-established in the literature that high-level tasks like 3D detection can achieve improved performance through multi-dataset training. This makes it particularly interesting why low-level tasks—previously claimed not to benefit from such training—now demonstrate clear gains. The shift deserves further explanation.

3. The paper unifies frame rate to 10 Hz (interpolation) and applies simple augmentations; tighter ablations on these standardization choices could clarify which components matter most.

**Questions:**

How do rain/occlusion/noise distributions affect failures? Could adaptive confidence weighting or uncertainty-aware post-processing mitigate the artifacts shown?

---

### Note · Authors · 2025-11-12

I have read and agree with the venue's withdrawal policy on behalf of myself and my co-authors.